# Near Missed Case of Occupational Pleural Malignant Mesothelioma, a Case Report and Latest Therapeutic Options

**DOI:** 10.3390/ijerph192214763

**Published:** 2022-11-10

**Authors:** Claudia-Mariana Handra, Marinela Chirila, Raluca-Andreea Smarandescu, Isabel Ghita

**Affiliations:** 1Occupational Health Department, Carol Davila University of Medicine and Pharmacy, 050474 Bucharest, Romania; 2Faculty of Pharmacy, Titu Maiorescu University, 040441 Bucharest, Romania; 3Pharmacy and Pharmacology Department, Carol Davila University of Medicine and Pharmacy, 050474 Bucharest, Romania

**Keywords:** pleural malignant mesothelioma, asbestos, occupational exposure

## Abstract

Asbestos use started to be gradually banned in Europe from 1991 onwards, and there are currently strict occupational exposure limits for asbestos. However, malignant mesothelioma has a long latency time (in some cases up to 50–60 years), so the risks related to asbestos exposure should not be forgotten. Considering the increased risk of lung cancer following the inhalation of asbestos fibers, lifetime health monitoring should be considered in people occupationally exposed to asbestos, with an emphasis on the respiratory system. An assessment of their occupational history should be performed rigorously, especially in the areas with a history of asbestos production/use, as this is a key element for an early diagnosis and appropriate treatment. This case report presents a near-missed case of occupational pleural malignant mesothelioma. The latency time between the first asbestos exposure and the diagnosis of occupational pleural malignant mesothelioma was 49 years. The accurate diagnosis was made two years after the first symptoms appeared.

## 1. Introduction

Malignant mesothelioma is a rare and aggressive form of cancer, with a world standardized incidence rate (WSIR) per 100,000 persons in Europe of 1.7 for males and 0.4 for females. Malignant mesothelioma is most frequently located at the pleural level, with approximately 60–70% of cases, followed by the peritoneal (30%) and pericardial (1–2%) localization [1]. The incidence of the disease varies based on the geographical location of the asbestos exploitations. In Japan, there were 7 cases/million of malignant mesothelioma compared to Australia, where there were 40 cases/million [2]. It is estimated that 43,000 patients diagnosed with malignant pleural mesothelioma die annually, of which over 10,000 cases are registered in North America, Western Europe, Australia, and Japan [3].

The etiology of malignant mesothelioma is related to exposure to mineral fibers, especially asbestos fibers [4]. Asbestos is a generic name used to describe a group of natural silicates with a fibrous structure that is used in a wide range of products such as building materials (asbestos-cement plates), brake pads, and ferrodos for cars, electrical components, vinyl, tiles, textiles, even cigarette filters, flame retardants, and insulators. In the last century, two types of asbestos were commonly used at the industrial level: amphiboles which include crocidolite, amosite, and serpentine (also named chrysotile) [5]. Currently, more than 60 countries have banned the import and use of asbestos. However, there are still countries, such as Russia, Kazakhstan, China, Brazil, and Zimbabwe that continue to produce and export asbestos [6,7]. All types of asbestos are currently considered fibrogenic and carcinogenic [5], and there is no safe limit for asbestos exposure [8]. Human mesothelial cells are very sensitive to the deleterious effects of asbestos which induce the production of reactive oxygen and nitrogen species, the release of TNF-α, growth factors, and other cytokines, and DNA damage [9]. The results of various studies showed that asbestos fibers reach the pleural space where they accumulate, with a subsequent potential risk of inducing pleural fibrosis and tumors [10].

The period of latency between exposure and the appearance of the disease is, on average, 40 years [11], and thus, the population at high risk should be monitored even after the cessation of exposure. The purpose of this report is to present a nearly missed case of occupational pleural malignant mesothelioma that resulted from a lack of measures to prevent asbestos exposure in the workplace atmosphere; due to a very long latency, the occupational history was overlooked. Attending physicians often are not acquainted with the indicators of the condition, which in turn, can result in a late diagnosis.

## 2. Case Presentation

A 68-year-old non-smoking female patient with a history of essential hypertension, dyslipidemia, and hypothyroidism, was admitted in February 2022 to the Colentina Occupational Medicine Clinic due to dyspnea, asthenia, loss of appetite, weight loss, and pain in the left thoracic area. Clinical examination revealed severe skin and mucous membrane pallor, a postoperative scar in the left thoracic region, pleural rubs in the left pulmonary base, dullness in the lower base of the right thoracic area, an abolished vesicular murmur in the right pulmonary base, 92% oxygen saturation at rest.

In 2019, the patient was admitted to a local hospital in the respiratory department for dyspnea on exertion during moderate physical activity. A clinical evaluation and thoracic computed tomography (CT) revealed the presence of pleural effusion in the left hemithorax (Figure 1a). The result of the biochemical and cytological analysis of the pleural fluid showed a lymphocyte-predominant exudate. The patient was diagnosed with pleurisy of an unspecified etiology, and antibiotic and anti-inflammatory treatment was recommended for 7 days. The evolution was favorable, and the pleural fluid was reabsorbed.

Between 2019 and 2021, the dyspnea progressively accentuated, and it was accompanied by chest pain and weight loss. 

In September 2021, the patient was admitted to a thoracic surgery department. A thoracic CT showed pleural effusion (Figure 1b). Thoracentesis and a pleural biopsy were performed, and the patient was diagnosed with “Diffuse malignant mesothelioma with epithelioid elements”.

In October 2021, a left thoracotomy and decortication of the pleura were performed. The histopathological examination showed diffuse biphasic mesothelioma (with 60% sarcomatous elements and 40% epithelioid elements) with pulmonary invasion. In the epithelioid areas, the cells were arranged in tubulopapillary and trabecular structures, while in the sarcomatous areas, the cells had a transitional morphology. Areas of necrosis, moderate atypia, and mitosis were also identified. Immunohistochemical tests were carried out, and the results were positive for Calretinin, Wilms tumor 1, and D 24–0 supporting the diagnosis of mesothelioma. The results of the immunohistochemical tests are presented in Table 1.

To assess the extension of the lesions, a positron emission tomography (PET-CT) was performed. The results showed the thickening of the left pleura from the top to the base. No metastases or invasion of the chest wall, mediastinum, or diaphragm were noted (Figure 2).

The patient completed five sessions of radiotherapy with modular intensity and image-guided radiotherapy and followed chemotherapy (cisplatin 100 mg and pemetrexed 750 mg).

In January 2022, the patient’s condition worsened, and the patient reported dyspnea at rest and orthopnea. A thoracic CT showed right pleural effusion (Figure 1c). Thoracocentesis was performed, and the analysis of the pleural fluid revealed the presence of lymphocytes and mesothelial cells. The cultures for mycobacterium tuberculosis were negative. Following antibiotic and anti-inflammatory treatments, a slight improvement in dyspnea was noticed.

A detailed occupational history was taken on admission to the Occupational Medicine Clinic in February 2022. The patient completed a total of 36 years of work, with a history of 1 year and 10 months of occupational exposure to asbestos in the period from 1973 to 1975. During that period, the patient worked in a factory that produced asbestos-cement plates using chrysotile in the production process. The patient worked as a quality control specialist, and her duties were to check if the thickness of the asbestos-cement boards complied with the technical specifications. From 1975, she worked for 34 years as a teacher in a middle school. The patient retired in 2014.

In February 2022, the patient was diagnosed with malignant pleural mesothelioma following occupational exposure to asbestos and right pleurisy.

The evolution was not favorable contralateral pleurisy appeared with the aggravation of the general condition. In June 2022, the patient started radiotherapy treatment.

## 3. Discussion

Because the use of asbestos has been banned in Europe for many years now, the exposure to asbestos and its associated risks are many times overlooked by doctors and patients; however, the risks associated with asbestos exposure should not be neglected because malignant mesothelioma occurs after a period of over 40 years from exposure.

Malignant mesothelioma is a rare and aggressive form of cancer, with a higher incidence rate in males compared to females [1]. Mesothelioma develops in the mesothelial cells of the pleura, peritoneum, pericardium, and testicular tunica vaginalis. The ethology is related to exposure to mineral fibers, especially asbestos fibers [4].

Pleural malignant mesothelioma is an aggressive type of cancer of the pleural surface associated with previous asbestos exposure, with a very long latency time and poor prognosis [12]. Clinical manifestations of malignant pleural mesothelioma begin insidiously and are often nonspecific, which causes the diagnosis to be delayed after successive evaluations of the patients with pleurisy [3]. Pleural effusion may be asymptomatic or with minimal manifestations and reabsorb completely or may be recurrent, hemorrhagic, with large amounts of fluid [13,14]. In the case of malignant pleural mesothelioma, cytological examinations may show the presence of atypical cells.

Regarding imaging investigations, chest X-rays are usually the first investigation to be performed, and it shows unilateral pleural effusion in 30–80% of cases. Computed tomography with contrast material is mandatory even for the initial evaluation and plays a leading role in staging, tracking the evolution, and establishing the therapeutic protocol [15]. PET-CT is a complementary investigation needed for differential diagnosis of benign pleural disorders as it can provide additional information on the metabolic function of tissues. Typically, in the case of mesothelioma, areas of abnormal pleural thickening show a higher consumption of the radiotracer as compared to benign pleural diseases [13].

A certain diagnosis is established by pleural biopsy under tomographic guidance or by biopsy performed with video-assisted thoracotomy [15].

From an anatomopathological point of view, pleural mesothelioma is a heterogeneous tumor including three main histological types: epithelioid (60–80%), sarcomatoid (<10%), and biphasic or mixed (10–15%), and others less common desmoplastic, with small cells and lymphocyticoid. Identifying the histopathological type is a useful tool for therapeutic conduct and provides information on survival analysis. The epithelioid form is less aggressive compared to the sarcomatoid form, as it is associated with higher sensitivity, a better response to chemotherapy, and a better survival rate compared to sarcomatoid or biphasic types [16]. In the present case, the patient was diagnosed with diffuse biphasic malignant mesothelioma, which is known to have an aggressive evolution, and was confirmed by the damage to the contralateral pleura shortly after starting the chemotherapeutic treatment.

Immunohistochemical tests have an essential role in positive and differential diagnoses. Typically, BAP1 and p16FIH are negative in malignant mesothelioma, and they are used in differential diagnosis to identify benign diseases [11]. On the other hand, the differential diagnosis with pulmonary adenocarcinoma is supported by the association of at least two positive mesothelial immunohistochemical markers (calretinin, cytokeratine 5/6, Wilms tumor 1, D 24–0) with two negative markers for adenocarcinoma (TTF1, CEA, BerEP4) [13]. In the present case, the immunohistochemical examination revealed three positive markers for mesothelioma: Calretinin, Wilms tumor 1, and D 24–0.

Currently, the treatment of malignant mesothelioma includes chemotherapy, radiotherapy, and the surgical decortication of the pleura. First-line chemotherapy consists of a combination of cisplatin and pemetrexed or raltitrexed. The cisplatin and pemetrexed combination was reported to be associated with a median survival of 13.3 months, compared with 12.7 months which were reported following the administration of cisplatin alone [17]. Patients treated with bevacizumab, an anti-vascular endothelial growth factor (anti-VEGF), showed an increase in overall survival rate according to several studies conducted on the matter [18]. Immunotherapy had revolutionary results in the treatment of lung cancer and melanoma, two types of cancer for which treatment options have been also limited until now [19]. Recent research has shown the greater efficacy of immunotherapeutic treatment in the form of non-epithelioid mesothelioma, and the use of nivolumab in combination with ipilimumab as a first-line treatment has been approved in various countries [15,20]. Unfortunately, the most frequent histological type of mesothelioma is the epithelioid one (60–80%). Other immunotherapy combinations are under evaluation, such as cisplatin, pemetrexed, and durvalumab or carboplatin, bevacizumab, and atezolizumab [20]. The results of a phase I study showed that tremelimumab, a fully human monoclonal antibody, was safe and well tolerated in monotherapy and combination therapy with durvalumab in Japanese patients with advanced cancer or malignant mesothelioma [21]. In addition, the combination of nivolumab/ipilimumab has been approved by the Food and Drug Administration as a first-line treatment for malignant pleural mesothelioma, and it is expected that the results obtained will improve therapies with checkpoint inhibitors [18].

The surgical treatment of pleural mesothelioma remains controversial. The selection of the patient is difficult due to the advanced stage of the disease, older age, and comorbidities. In addition, the surgery on the pleura must be radical. The form of the sarcomatous tumor is excluded from the surgical treatment, and the therapeutic opinions in the case of biphasic mesothelioma are different [15].

The third treatment method, postoperative radiation therapy, is performed in doses of up to 54–60 Gray, and it is administered with new techniques such as intensity-modulated radiation therapy (IMRT) and image-guided radiation therapy (IGRT) [19].

Even if the current treatment options for pleural mesothelioma seem to be encouraging, the survival rate and the quality of life are not substantially improved yet. Unfortunately, there are no clinical signs in the early phase of this deadly disease, so diagnosis is made in a late stage. The currently available chemotherapy or surgery options are not providing a significant improvement in survival rate if applied in the early stages of the disease [22,23,24]. Therefore, the screening of formerly asbestos-exposed workers with the aim of identifying early-stage mesothelioma would not assure better survival rates with the current treatment options. However, there are some promising results from the studies on immunotherapy that are undergoing. Once available, the results of these studies might help to understand if these treatments provide better survival rates in patients with early-stage mesothelioma; if so, monitoring the asbestos-exposed workers by means of a screening program would be beneficial. More research is, however, needed.

This case report presents an occupational pleural malignant mesothelioma in a patient exposed to asbestos for 1 year and 10 months. For the present case, the latency time between the first asbestos exposure and the diagnosis of occupational pleural malignant mesothelioma was 49 years. This long latency could be attributed to the short (1 year and 10 months) and/or low exposure (as the patient was doing quality control checks of the finished products). The accurate diagnosis was made 2 years after the first symptoms appeared; the late diagnosis can be explained by the lack of doctor experience in this rare disease. The treatment included the decortication of the left pleura, radiotherapy, and chemotherapy (with cisplatin and pemetrexed). Unfortunately, the evolution was not favorable. After the third course of chemotherapy, contralateral pleurisy appeared with the aggravation of the general condition.

The asbestos exposure of this patient happened in the 70s in a Romanian factory with approximately 300 employees. The work was conducted in an asbestos processing plant, where all operations were carried out in a single large room. The protective equipment did not ensure respiratory protection, so exposure to asbestos was possible even for the personnel from the quality control department. No determinations of asbestos fibers were made at the workplace during that period. The health monitoring of these asbestos-exposed workers was carried out yearly, as requested by the local legislation, by the occupational medicine physician but only during asbestos exposure; no health monitoring was conducted after retirement or after changing occupation. There are no data available related to post-exposure medical follow-ups/surveillance of the workers from this factory.

Statistical data related to workers exposed to asbestos in Romania are insufficient. The data provided by the Public Health Institute show that 1200 people exposed to asbestos were registered in Romania in 2006.

In January 2007, the production, sale, and the use of asbestos was banned in Romania. However, the current national guidelines allow the treatment and disposal of products which result from the demolition and removal of asbestos [25]. A guideline for post- professional exposure to carcinogenic agents, including asbestos, is currently under preparation in Romania.

The present case reiterates the information provided by the literature data that mesothelioma is related to occupational exposure to asbestos and that it has a long latency time, sometimes longer than 40 years.

The latency time was longer when the exposure duration or intensity to asbestos was short. Therefore, lifetime health monitoring with screening for mesothelioma and other asbestos-related diseases should be considered in people occupationally exposed to asbestos. In addition, rigorous occupational health records should be taken, especially in areas with a history of asbestos production/use.

Insidious debut and nonspecific clinical manifestations of malignant pleural mesothelioma contribute to the late diagnosis associated with the advanced stage of the disease and low survival rate.

## 4. Conclusions

In the present case, the patient was not aware of the risk related to asbestos exposure, and no health monitoring was conducted. Asbestos exposure was overlooked for 2 years. The treatment included the decortication of the left pleura, radiotherapy, and chemotherapy (with cisplatin and pemetrexed). The standard therapeutic protocols were applied, and none of the newly approved or novel medicines were used. Unfortunately, the evolution was not favorable. After the third course of chemotherapy, contralateral pleurisy appeared with the aggravation of the general condition.

## 5. Take-Home Message

Occupational asbestos exposure may be overlooked, especially in cases of short-term exposure and long latency until the onset of the malignant mesothelioma.There is a need to increase awareness of the risks associated with asbestos exposure among current and former workers involved with asbestos-related work.Lifetime monitoring of the persons exposed to asbestos with screening for mesothelioma and other asbestos-related diseases should be recommended.A rigorous occupational history is essential for an early accurate diagnosis that might be associated with a better prognosis in the case of malignant pleural mesothelioma.

## Figures and Tables

**Figure 1 ijerph-19-14763-f001:**
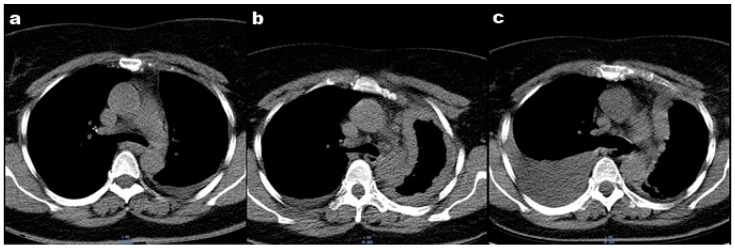
Thoracic computed tomography: (**a**) Thoracic CT performed in 2019 showing pleural effusion in the left hemithorax; (**b**) Thoracic CT performed in 2021 showing pleural effusion in the left hemithorax; (**c**) Thoracic CT performed in 2022 showing right pleural effusion.

**Figure 2 ijerph-19-14763-f002:**
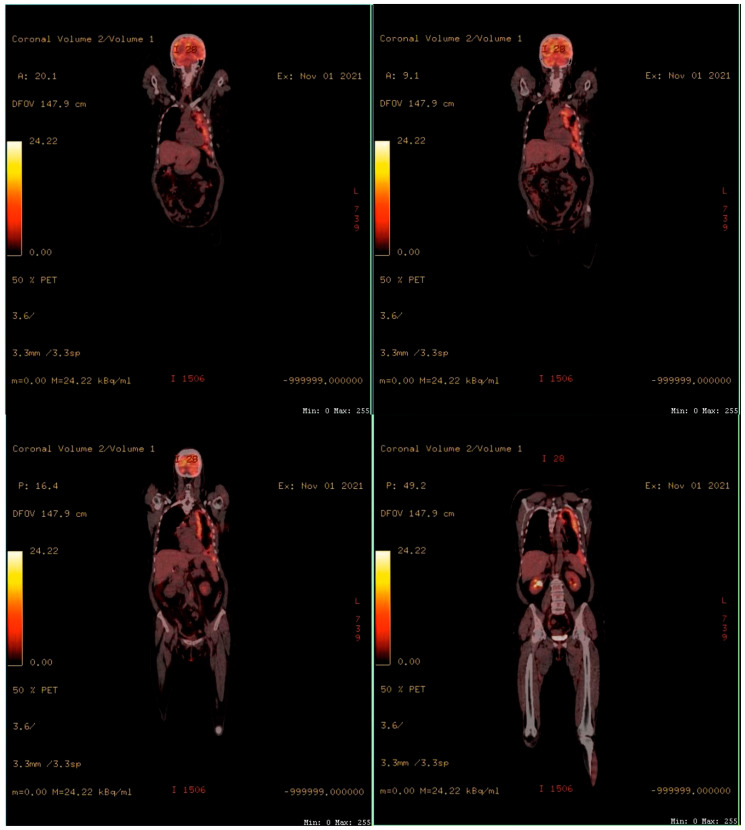
PET-CT scans showing thickening of the left pleura from top to base. No metastases or invasion of the chest wall, mediastinum, or diaphragm.

**Table 1 ijerph-19-14763-t001:** Immunohistochemical tests results.

Antibody	Findings
BAP1 (C-4)	No loss of nuclear expression
BER-EP4 (Ber-EP4)D	Negative
Claudin 4 (3E2C1)	Negative
D2–40 (D2–40)D	Positive
Calretinin (DAK-Calret 1)D	Positive
WT1 (6F-H2)	Positive

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
