# Peer review of "Near Missed Case of Occupational Pleural Malignant Mesothelioma, a Case Report and Latest Therapeutic Options"

_ijerph, 2022, doi:10.3390/ijerph192214763_

Round 1
Reviewer 1 Report
The introduction provides insufficient background.
Therefore, I recommend the improvement of discussion, to include more references, more discussion on prevention, monitoring and treatment aspects of the disease and the link with professional exposure.
It would be interesting to discuss the occupation she had in the period 1973-1975 and what responsibilities she had in that factory.
Is there another case known or is there evidence of the pathology of the colleagues from that time?
What does the monitoring of people in contact with asbestos fibers include and how often is it done in Romania after asbestos exposure?
I recommend the inclusion in the presentation of the case the images of the histopathological examination and immunohistochemical tests sections.
Author Response
Dear Sir or Madam,
Thank you for reviewing our manuscript and for the recommandations. The paper was amended considering all points raised.
Please see the attachment.
With kind regards,
Claudia Handra

Reviewer 2 Report
It is a case report mainly described from the clinical side. Just a brief reference to a low and distant exposure in an asbestos cement company (47 years ago). But well described from a diagnostic-therapeutic point of view. In my experience when a precise diagnosis is late depends on a poor experience of the diagnostic centre on this particular disease which remains rare, although important from the social point of view because mainly caused by the poor attention to occupational prevention in workplaces. Furthermore, it’s a disease which becomes fatal for the patients and this is the main reason why we need to keep discussing about it through publications and above all: research.
Since a couple of years ago the screening of the past exposed workers was useful only to diagnose lung cancers in early stage and therefore hoping in a good prognosis after a good surgery. Unfortunately, an early stage diagnosis of mesothelioma didn’t allow the same procedure because the survival of the patients didn’t show any advantage of the chemotherapy or surgery performed in early stages.
Now the research on immunotherapy treatments is giving some positive effects. There are some studies in progress and soon they will be hopefully published.
So, if the positive effects on survival of immunotherapy treatments on patients affected by early stage mesothelioma will be confirmed, a screening program of the former exposed workers would be appreciated. Unfortunately, the preliminary results of immunotherapy treatment say that it works only for “non epithelial” histological types, while the majority of mesotheliomas have this histological characterization.
Authors are invited to discuss more deeply this crucial topic.
The claim that longer latencies are due to short or low exposures is appreciable but the Authors should report a more detailed work history before assuming this statement. One year and ten months appear to be a short period compared to those who spend their whole life doing the same job in asbestos cement manufacturing, but it is essential to know, even just an estimate, of the intensity of exposure in that factory. After all, the authors claim to be in possession of a detailed work history “A detailed occupational history was taken on admission to the Occupational Medi-81 cine Clinic in February 2022”. It is a matter of fact that the length of latencies has a positive trend because heavy asbestos exposures happened during the XX° century and then stopped or had an important reduction in intensity.
Authors are also invited to describe more precisely in terms of periods when the production of asbestos cement ceased and give an estimate of the number of former asbestos exposed workers in their country.
The article can be accepted with some revisions and integrations.
Author Response
Dear sir or Madam,
Thank you for reviewing our manuscript and for all the recommandations. Indeed, we personally lack experience with such a rare disease and thus your comments are very valuable to us.
Please see the attachment.
With kind regards,
Claudia Handra
